# A Million-Cow Genome-Wide Association Study of Three Fertility Traits in U.S. Holstein Cows

**DOI:** 10.3390/ijms241310496

**Published:** 2023-06-22

**Authors:** Zuoxiang Liang, Dzianis Prakapenka, Paul M. VanRaden, Jicai Jiang, Li Ma, Yang Da

**Affiliations:** 1Department of Animal Science, University of Minnesota, Saint Paul, MN 55108, USA; zliang@umn.edu (Z.L.); praka032@umn.edu (D.P.); 2Animal Genomics and Improvement Laboratory, Agricultureal Research Service, United States Department of Agriculture, Beltsville, MD 20705, USA; paul.vanraden@usda.gov; 3Department of Animal Science, North Carolina State University, Raleigh, NC 27695, USA; jicai_jiang@ncsu.edu; 4Department of Animal and Avian Sciences, University of Maryland, College Park, MD 20742, USA; lima@umd.edu

**Keywords:** GWAS, fertility, daughter pregnancy rate, cow conception rate, heifer conception rate, SNP, Holstein cow

## Abstract

A genome-wide association study (GWAS) of the daughter pregnancy rate (DPR), cow conception rate (CCR), and heifer conception rate (HCR) using 1,001,374–1,194,736 first-lactation Holstein cows and 75,140–75,295 SNPs identified 7567, 3798, and 726 additive effects, as well as 22, 27, and 25 dominance effects for DPR, CCR, and HCR, respectively, with log_10_(1/p) > 8. Most of these effects were new effects, and some new effects were in or near genes known to affect reproduction including *GNRHR*, *SHBG*, and *ESR1*, and a gene cluster of pregnancy-associated glycoproteins. The confirmed effects included those in or near the *SLC4A4-GC-NPFFR2* and *AFF1* regions of Chr06 and the *KALRN* region of Chr01. Eleven SNPs in the *CEBPG-PEPD-CHST8* region of Chr18, the *AFF1-KLHL8* region of Chr06, and the *CCDC14-KALRN* region of Chr01 with sharply negative allelic effects and dominance values for the recessive homozygous genotypes were recommended for heifer culling. Two SNPs in and near the *AGMO* region of Chr04 that were sharply negative for HCR and age at first calving, but slightly positive for the yield traits could also be considered for heifer culling. The results from this study provided new evidence and understanding about the genetic variants and genome regions affecting the three fertility traits in U.S. Holstein cows.

## 1. Introduction

The U.S. Holstein genomic evaluation included three female fertility traits, daughter pregnancy rate (DPR) as the percentage of cows that become pregnant during each 21 d period [1], and cow conception rate (CCR) and heifer conception rate (HCR) as measures of the conception ability of cows and heifers [2,3]. These fertility traits all have low heritability [4,5,6] with large random variations, which made identifying and understanding the genetic factors affecting these traits challenging. For the same three fertility traits in U.S. Holstein cattle, several GWAS were reported, including studies using 27,214 Holstein bulls [7], 24,000 Holstein bulls and 36,000 cows [4], and a large-scale GWAS using 186,188–269,158 Holstein cows [8]. These studies have accumulated some consensus for genetic effects affecting the three fertility traits, including effects in the *GC* and *KALRN* genes for DPR and CCR [7,8] and in the *AFF1* gene for HCR [4,8]. The sample sizes of 186,188–269,158 Holstein cows for the large-scale GWAS of DPR, CCR, and HCR were unprecedentedly large in 2019, but the numbers of significant effects detected for the three fertility traits were far fewer than those for the five production traits: 15–360 additive effects and 1–2 dominance effects for each of the three fertility traits compared to 9803–15,215 additive effects and 24–152 dominance effects for each of the production traits [8]. Although the numbers of the true effects for those fertility and production traits were unknown, the much smaller numbers of fertility effects likely were due to the small sizes of the fertility effects relative to the large random variations. Under this assumption, increased sample sizes should detect more genetic effects affecting the fertility traits and provide more reliable estimates of the genetic effects. Since the 2019 GWAS, the sample sizes of U.S. Holstein cows increased rapidly. By the end of 2022, the number of cows that could be used for GWAS for each of the three fertility traits surpassed one-million. Such large sample sizes should provide much greater statistical power for detecting more fertility effects assuming the numbers of the true effects were substantially greater than previously detected, for producing more reliable estimates of the genetic effects and for distinguishing between the likely true rare genetic effects from spurious rare genetic effects. Because of these advantages, a large-sample GWAS is a powerful tool for understanding the genome functions affecting phenotypes by providing high-confidence targets of genomic variants, genes, and regions affecting quantitative traits. Using the Holstein million-cow resources, we conducted a GWAS for the three fertility traits of U.S. Holstein cows to obtain a new understanding of the genetic factors affecting the three fertility traits.

## 2. Results and Discussion

The GWAS detected 7576, 3798, and 726 additive effects and 22, 27, and 25 dominance effects with log_10_(1/p) > 8 for DPR, CCR, and HCR, respectively, for a total of 12,100 additive effects and 74 dominance effects for the three fertility traits. These effects were considerably more than the 487 additive effects and 5 dominance effects for the same three fertility traits reported in the 2019 study [8]. The new effects included some effects involving genes known to affect reproduction and some rare, but very negative effects and confirmed some of the previously detected effects. The additive effects each with a small effect, except with a few sharply negative additive effects, pointed to the polygenetic nature of the fertility traits. The number of significant dominance effects (74 effects) was much smaller than the number of additive effects (12,100 effects), but some of the dominance effects involved rare genotypes with sharply negative dominance values. The details of the GWAS results are provided in the Appendix A, including the top-1000 additive effects for DPR (Appendix A) and CCR (Appendix A), all 726 significant additive effects for HCR (Appendix A), and all 74 significant dominance effects of the three fertility traits (Appendix A), whereas the main text focused on the most-significant chromosome regions and effects. The statistical significance measured by the log_10_(1/p) from the *t*-test of Equation (5) is an increasing function of the size of the additive effect (α defined by Equation (10)) for the additive test or the size of the dominance effect (δ defined by Equation (12)) for the dominance test and is an increasing function of the minor allele frequency for the additive test and minor genotypic frequency for the dominance test. Therefore, an SNP with a rare allele or genotype, but a large additive or dominance effect could be highly significant, and an SNP with a small additive or dominance effect, but a large minor allele or genotypic frequency could also be highly significant if the sample size is sufficiently large such as the million-cow samples in this study. The results of this study had examples of both cases. For the same effect size, the alternative alleles and genotypes of the same SNP may have drastically different impacts on the phenotype. Therefore, the presentation and discussion of the results will cover the statistical significance, the effect size, the allelic effects and dominance values, and the allele and genotypic frequencies associated with the allelic effects and dominance values.

### 2.1. Additive Effects for DPR

Two chromosome regions had highly significant additive effects, the 83.44–93.83 Mb region of Chr06 with peak effects in or near *GC*, *NPFFR2*, and *SLC4A4* and the 43.59–51.11 Mb region of Chr18 with peak effects in or near *PAK4*, *DEDD2*, *POU2F2*, *CEBPG*, *PEPD*, and *SIPA1L3*. Other chromosome regions with similar statistical significance included the *IGSF5-DSCAM* and *KALRN* regions of Chr01, the *AMN1* region of Chr05, the *PSMC3* region of Chr09, the *SHBG-ATP1B2* region of Chr19, and the *LOC104972343* region of Chr23 (Figure 1a). Among the top-1000 most-significant SNP effects (Appendix A), negative allelic effects on average had larger sizes (average −0.29) than positive allelic effects (average 0.20), with chromosomes 1 and 18 having sharply lower additive effects than those of other chromosomes (Figure 1b).

The 83.44–93.83 Mb region of Chr06 (10.4 Mb in size) had a large cluster of highly significant additive effects (Figure 1c, Appendix A). In this large region, positive and negative effects on average had similar sizes, an average of 0.23 for positive effects and −0.22 for negative effects. The three SNPs in *GC* (GC vitamin D binding protein), *CSN1S2* (casein alpha-S2) and *CSN3* (casein kappa) each had the largest positive effects, and the SNP effect in *GNRHR* (gonadotropin-releasing hormone receptor) was an example of symmetric effects, where the positive and negative effects of the SNP had similar sizes (Figure 1c, Appendix A). The known biological functions of these four genes should all be relevant to dairy cows.

Within this large 10.4 Mb region, the 86.79–87.32 region (0.53 Mb in size) containing *GC*, *NPFFR2*, and *SLC4A4* had eleven of the top-twenty significant effects (Table 1). In this 0.53 Mb region, negative effects on average had larger sizes than the positive effects, an average of −0.32 for negative effects and 0.29 for positive effects. Two SNPs between *SLC4A4-GC*, one SNP between GC-*NPFFR2*, and one SNP in *NPFFR2* had the largest negative effects and were the most-significant effects among all SNPs (Figure 1c, Appendix A, Appendix B Figure A1). The #1 significant effect (*rs110434046*) was between GC and *NPFFR2*, about 220 Kb downstream of *GC* (Table 1, Appendix B Figure A1). 

The 86.79–87.32 region had two examples indicating the power of large samples for fine mapping to distinguish the relative importance of two closely linked loci to the phenotype. The two SNPs in *NPFFR2*, *rs109034709* and *rs137844449*, were only 1062 bp apart, but had very different statistical significance and effect sizes, log_10_(1/p) = 94.27 (#2 ranking) and |α| = 0.71 for *rs109034709* and log_10_(1/p) = 32.28 (#286 ranking) and |α| = 0.41 for *rs137844449* (Table 1, Appendix B Figure A1). This example showed that the sample size of 1,194,736 cows for DPR was able to distinguish the additive effects of two SNPs as close as 1062 bp apart, noting that the two SNPs had similar minor allele frequencies, 0.364 for *rs109034709* and 0.375 for *rs137844449* (Table 1 and Appendix A). Another two closely linked SNPs in the same region also had very different log_10_(1/p) values (Appendix B Figure A1).

The additive effects of Chr18 among the top-1000 effects were in the 43.10–59.75 Mb region of the chromosome (Figure 1d, Appendix A). This Chr18 region had seven SNPs with the top-20 effects (ranking #8-#19) in or near *PAK4*, *DEDD2*, *POU2F2*, *CEBPG*, *PEPD*, *CHST8*, and *SIPA1L3* (Table 1). Among these seven SNPs, three SNPs in or near *CEBPG*, *PEPD*, and *CHST8* had some of the most-negative effects, which will be discussed later along with the dominance effects of those SNPs.

The additive effects of Chr01 among the top-1000 effects were mostly in two regions: the 68.04–70.7b Mb and 134.36–141.05 Mb (Figure 1e). In the 68.04–70.7b Mb region, there were two SNPs in *KALRN* and one SNP in *CCDC14*. These sharply negative allelic effects along with those in or near *CEBPG*, *PEPD*, and *CHST8* of Chr18 had a common feature: the alternative alleles of the SNP behaved like neutral alleles with slightly positive allelic effects above the “0” line (Figure 1d). The 134.36–141.05 Mb had the most-significant effect of Chr01 in *IGSF5*. This SNP along with the SNP in *PRDM15* had some of the highest positive allelic effects of this chromosome (Figure 1e), along with the positive effects in *GC*, *CSN1S2*, and *CSN3* of Chr06 (Figure 1c).

The additive effects of Chr05 among the top-1000 effects were in the 24.82–103.67 Mb region (Figure 1f, Appendix A) with the most-significant effects in *AMN1* and *SLC38A4* and the most-negative effects in *C2CD5* and near *KRAS*.

The most-significant additive effect of Chr19 (#46 among all significant additive effects) was 5139 bp downstream of *SHBG* (sex-hormone-binding globulin), and this SNP also had a highly significant effect on AFC [9].

### 2.2. Dominance Effects for DPR

DPR had 22 significant dominance effects with log_10_(1/p) > 8 (Appendix A). The dominance tests showed that only four chromosomes had significant dominance effects for DPR, Chr18, Chr06, Chr01, and Chr09 (Table 2, Figure 2a–e). The *CEBPG*-*PEPD*-*CHST8* region of Chr18 had the most-significant dominance effects and the most-negative dominance values of the homozygous recessive genotypes (d_RR), followed by the *AFF1 KLHL8* region of Chr06 and the *CCDC14*-*KALRN* region of Chr01. The dominance values of the homozygous recessive genotypes of Chr09 were also negative (Figure 2e), but not as negative as those of Chr18, Chr06, and Chr01. The most-significant dominance effect of Chr04 (Figure 2f) was statistically insignificant for DPR, but this effect was more negative than those of Chr09 and was highly significant for HCR to be described later. 

The number of significant dominance effects was much smaller than those of the additive effects, but the size of each dominance effect generally was much larger than that of the additive effects. Among the 22 significant dominance effects (Appendix A), 13 dominance effects had effect sizes greater than the largest additive effect, 1.86–4.81-times as large as the largest additive effect, and all these 13 dominance effects had rare homozygous recessive genotypes with genotypic frequencies of 0.3–0.6% (Table 2).

### 2.3. Additive Effects for CCR

The additive effects for CCR (Appendix A) overlapped by 60.5% those of DPR (Appendix A) among the top-1000 additive effects of each trait. The highly significant additive effects for CCR (Figure 3a–f) mostly overlapped with the highly significant effects for DPR (Figure 1a–f) with some differences. The same SNP (*rs110434046*) of Chr06 had the most-significant additive effect for both CCR (Appendix A) and DPR (Table 1). A major difference was that Chr03 was more significant for CCR (Figure 3a) than for DPR (Figure 1a). The #2 most-significant effect for CCR among all SNPs was on Chr03 (Appendix A), and this effect was #59 for DPR (Appendix A). The additive effects of Chr03 for CCR covered a large distance of Chr03 and was mostly in the region of 44–64 Mb, and most of the negative alleles had larger effect sizes than the positive effect sizes (Appendix C Figure A2). Another difference was that Chr01, Chr03, Chr05, Chr06, and Chr18 had similar statistical significance for the most-significant effects on each chromosome for CCR (Figure 3a), unlike DPR, where Chr06 and Chr18 were considerably more significant than the other chromosomes (Figure 1a). Other differences between CCR and DPR included different effects among the top-1000 effects (Appendix A) and the location shifts of significant effects, e.g., the most-significant Chr01 effect for CCR was downstream of *7SK* (Appendix A), whereas the most-significant Chr01 effect for DPR was in *IGFS5* (Table 1). Among the top-1000 most-significant SNP effects, negative allelic effects on average had larger sizes (average −0.29) than positive allelic effects (average 0.24), with chromosomes 1 and 18 having sharply lower additive effects than those of other chromosomes (Figure 3c,d).

### 2.4. Dominance Effects for CCR

CCR had 27 significant dominance effects with log_10_(1/p) > 8 (Appendix A), and 19 of these 22 dominance effects overlapped 19 of the 22 dominance effects for DPR. The most-significant dominance effects for CCR were on Chr18, Chr06, Chr01, and Chr09 (Figure 4a–e), the same chromosomes with dominance effects for DPR. The dominance effect of CCR in estrogen receptor 1 (*ESR1*) of Chr09 (Figure 4e) was not shared by CCR and DPR. The dominance values of the three genotypes of this dominance effect had the smallest differences among all significant dominance effects for CCR. The reason for this dominance effect to be statistically significant was the large minor genotypic frequency of the three genotypes of 0.143 for the recessive genotype (the “22” genotype; Appendix A). This was an example of a significant effect with a small effect size, but a large minor genotypic frequency.

The *CEBPG*-*PEPD*-*CHST8* region of Chr18 had the most-significant and -negative dominance values of the homozygous recessive genotypes (Figure 4b), followed by the *AFF1-KLHL8* region of Chr06 (Figure 4c) and the *CCDC14*-*KALRN* region of Chr01 (Figure 4d). The dominance values of the homozygous recessive genotypes of Chr09 were also negative (Figure 4e), but not as negative as those of Chr18, Chr06, and Chr01. The most-significant dominance effect of Chr04 was statistically insignificant for DPR (Figure 2f), but this effect was more negative than those of Chr09 and was highly significant for HCR to be described later.

### 2.5. Additive Effects for HCR

HCR had 726 significant additive effects with log_10_(1/p) > 8 (Appendix A). The highly significant additive effects for HCR were in the *AFF1-KLHL8* region of Chr06, the *EPN2-GRAP* region of Chr19, and a chromosome region with multiple genes of pregnancy-associated glycoprotein (*PAG2-PAG7*, *PAG10*, *PAG11*) of Chr29 (Figure 5a). Among the 726 significant SNP effects, negative allelic effects on average had larger sizes (average −0.21) than positive allelic effects (average 0.17), with Chr01 and Chr06 having sharply lower additive effects than those of other chromosomes (Figure 5b–d). It was interesting that the sharply negative additive effects of Chr18 for CCR and DPR were not shared by HCR and the sharply negative dominance effects of Chr18 for CCR and DPR were also not shared by HCR, as to be shown. Some of the Chr19 effects on HCR also had highly significant effects on the age at first calving (AFC) [9], including SNPs in or near *NCOR1*, *ARHGAP44*, *HS3ST3A1-COX10*, and *SHBG* (Table 3 and Appendix A, Figure 5e). The pregnancy-associated glycoprotein genes of Chr29 (Figure 5f) were some of the genes detected in this study with known biological functions related to reproduction.

### 2.6. Common Effects for DPR, CCR, and HCR

The additive effects of the three fertility traits had 50 common SNPs (Appendix A). The overlapping effects were not ranked among the top-20 effects for any of the three traits. The highest-ranked effect was an SNP in *LOC100848253* for DPR (#28), upstream of *EYA4* for CCR (#44), and in *GLDN* for HCR (#33). The SNP in *ARHGAP44* (*rs110401500*) affecting all three traits was not ranked high (#295-#348, Appendix A), but had the most-significant effect for AFC [9]. Another SNP in *ARHGAP44* (*rs135166134*) had the #13 effect for HCR. These results of *ARHGAP44*’s involvement in the three fertility traits and AFC indicated that *ARHGAP44* had an important role in all these four reproductive traits. The SNP downstream of *SHBG* (*rs111004845*) was another interesting example of an SNP effect affecting all four reproduction traits, noting that the 80K SNP data in this study did not have any SNP in *SHBG*. The effects of this SNP for the three traits were ranked #37 for HCR, #46 for DPR, and #186 for CCR (Appendix A), but this SNP had the third-most-significant effect for AFC [9]. *SHBG* is one of the candidate genes with known biological functions affecting reproduction. This gene encodes a steroid-binding protein, and the encoded protein transports androgens and estrogens in the blood, binding each steroid molecule as a dimer formed from identical or nearly identical monomers [10]. Two other candidate genes (*COX10* and *TTC19*) also had effects affecting all four reproductive traits. These candidate genes, *ARHGAP44*, *SHBG*, *COX10*, and *TTC19*, should be strong candidate genes for their effects on all four reproductive traits.

### 2.7. Dominance Effects for HCR

HCR had 25 significant dominance effects with log_10_(1/p) > 8 (Appendix A) in four chromosome regions: the 22.13–35.41 Mb region of Chr04, the 101.88–102.21 Mb region of Chr06, the 76.68–77.96 Mb region of Chr08, and the 48.90–4919 Mb region of Chr09 (Figure 6a–e, Appendix A). An SNP upstream of *AGMO* and an SNP in *AGMO* were sharply negative for HCR (Figure 6b), noting that these SNPs did not have significant effects for DPR (Figure 2f) and CCR (Figure 4f). HCR dominance effects in the 101.88–102.21 Mb region of Chr06 were sharply negative (Figure 4c), and these effects were from the same SNPs in *AFF1* and *KLHL8*, which also had sharply negative dominance values for DPR (Figure 2f) and CCR (Figure 4f). All the highly significant dominance effects had low frequencies for the recessive genotypes and generally had larger effect sizes than the additive effects (Table 3 and Table 4).

### 2.8. Elimination of Rare Negative Recessive Genotypes for Heifer Culling

Based on the results of the additive and dominance effects for DPR, CCR, and HCR, we recommend 11 SNPs with sharply negative dominance values for heifer culling by eliminating heifers with the homozygous recessive genotype for any of the 11 SNPs, plus two Chr05 SNPs previously recommended for heifer culling based on AFC, for a total of 13 SNPs (Table 5). Of these SNPs, the three Chr01 SNPs were selected based on the additive effects for DPR and CCR (Figure 1e and Figure 3c); the four SNPs of Chr06 were based on the dominance effects for DPR, CCR, and HCR (Figure 2c, Figure 4c and Figure 6c); the four Chr18 SNPs were based on the additive and dominance effects for DPR and CCR (Figure 1d, Figure 2b, Figure 3d and Figure 4b). In addition, two Chr04 SNPs were selected based on the dominance effects for HCR (Figure 6b). Among these 15 SNPs, the 4 Chr06 SNPs and the 2 Chr05 SNPs were also recommended for heifer culling for AFC [9]. The homozygous recessive genotype of each SNP affected 2240–7079 cows among the more than one-million cows in this study (Appendix A); the thirteen SNPs of Chr01, Chr06, Chr18, and Chr05 affected 17,649–20,704 cows or 1.61–1.76%; all 15 SNPs including the 2 Chr04 SNPs affected 22,182–26,254 cows or 2.20–2.22% of the more than one-million cows in this study (Table 5).

To evaluate the consequence of eliminating the homozygous recessive genotypes using these SNPs, we calculated the measure of “negative impact” (NI), which is the difference between the average of the homozygous recessive phenotypic values and the average of the phenotypic values of the other two genotypes (Equation (14)) for seven traits, the three fertility traits, AFC, and the three yield traits (Table 5). The NI values provided estimated how much the homozygous recessive genotype of each SNP affected the seven traits if a heifer was eliminated for having the homozygous recessive genotype. 

The results showed that the Chr01, Chr06, and Chr18 SNPs had the most-negative NI for DPR and CCR, whereas the Chr06 and Chr04 SNPs had the most-negative NI for HCR and AFC (Table 5). Only the two Chr04 SNPs had a minor positive NI for the three yield traits, and all the remaining thirteen SNPs on Chr01, Chr06, Chr18, and Chr05 had a negative NI for all seven traits, noting that a positive NI value for AFC means an older first calving age and is interpreted as a negative result. The four Chr06 SNPs recommended by this study for all three fertility traits and by the AFC study [9] had the most-negative NI for AFC, and the two Chr05 SNPs recommended for AFC had the most-negative NI for the three yield traits. Therefore, all these thirteen SNPs could be recommended for culling heifers with the homozygous recessive genotype for any of the thirteen SNPs, whereas the use of the two Chr04 SNPs for heifer culling should consider the minor losses in milk, fat, and protein yields. We limit this recommendation to heifer culling because more research is needed to recommend the 13–15 SNPs for bull culling.

### 2.9. Interpretation of Dominance Effects

Dominance effects can be interpreted using three measures: dominance values (Equation (13)), corrected genotypic means after removing pedigree additive (breeding) values from the original phenotypic values (Equations (1) and (4)), and the genotypic means of the corrected phenotypic values after removing the pedigree additive (breeding) values from the original phenotypic values (Equations (1) and (4)) and the genotypic means of the original phenotypic values without removing the pedigree additive values. Based on dominance values that removed the SNP additive effects, all dominance effects were positive overdominance effects in the sense the heterozygous dominance value was more positive than the dominance value of either homozygous genotype of each SNP (Figure 2, Figure 4 and Figure 6). 

Based on the genotypic means of the corrected phenotypic values that removed the pedigree additive (breeding) values and the original phenotypic values that did not remove the pedigree additive values, the dominance effects were nearly complete dominance or partial dominance effects (Figure 7). The Chr04 example for HCR (Figure 7a,b) was nearly complete dominance where the heterozygous genotypic mean was nearly the same as the genotypic mean of the homozygous dominance genotype, whereas the genotypic mean of the homozygous recessive genotype was sharply lower than those of the other two genotypes. The Chr06 example for CCR (Figure 7c,d) was between nearly complete dominance and partial dominance, where the heterozygous genotypic mean was close to the genotypic mean of the homozygous dominance genotype, but not as close as that of the Chr04 example for HCR (Figure 7a,b). The Chr01 example for CCR (Figure 7e,f) was partial dominance, where the heterozygous genotypic mean was farther apart from the genotypic mean of the homozygous dominance genotype than in the Chr04 example for HCR (Figure 7a,b) and the Chr06 example for CCR (Figure 7c,d). DPR dominance effects (Appendix D Figure A3) provided more examples of the partial dominance of the dominant allele.

In all the examples of Figure 7, the dominant allele behaved like a neutral allele with the genotypic mean of the dominant homozygous genotypes nearly overlapping the average of the genotypic means of all SNPs (µ_all_) for the corrected phenotypic values after removing the pedigree additive (breeding) values from the original phenotypic values (Figure 7a,c,d) or nearly overlapping the phenotypic mean of all cows without removing the pedigree additive (breeding) values (Figure 7b,d,f). The lack of deviation of the genotypic mean of each homozygous dominant genotype from the population mean was considered to have no effect on the phenotype by the dominant allele when in homozygous status. DPR examples of the dominance effects (Appendix D Figure A3) provided further evidence that all dominant alleles in this study were neutral alleles when in homozygous status.

The most-important finding of the dominance effects in this article along with the dominance effects for AFC in a previous study [9] was the existence of dominant and recessive alleles for quantitative traits in dairy cattle, where the recessive allele had a sharply negative effect on the phenotype when in homozygous status and the dominant allele neutralized most of the negative effect of the recessive allele when in heterozygous status and behaved like a neutral allele with no effect on the phenotype when in homozygous status.

### 2.10. Statistical Significance, Effect Size, and Frequency

This study used the statistical significance measured by the log_10_(1/p) from the *t*-test as the primary evidence of the genetic effect of the fertility traits, but also considered the size and frequency of each genetic effect, because statistical significance is affected by the size of the genetic effect being tested (Figure 8a) and the allele frequency of the SNP (Figure 8b) and because the effect size and frequency have practical implications. Figure 8a shows a clear increasing trend of the statistical significance as the size of the additive effect increases, except a small number of SNPs with large effects, but medium statistical significance due to the low MAF of those SNPs. Figure 8b shows the statistical significance had a trend to be high for the high MAF around 0.5 of the negative alleles, although exceptions existed due to different effect sizes. The relationship between statistical significance and the frequency of the positive allele is just the mirror image of Figure 8b. 

A high statistical significance could be due to a large genetic effect and a small MAF such as the significance of *rs42739334* in *KLHL8* (Table 3) or a small genetic effect with a high MAF such as *rs3423297865* in *ESR1* (Appendix A). An SNP with a large and low-frequency effect had a large impact on the phenotype, but affected a small number of cows, whereas an SNP with a small and high-frequency effect had a small impact on the phenotype, but affected many cows. Among the positive and negative allelic effects, most effects had small sizes (Figure 8c,d), indicating the polygenetic nature of DPR with many genetic effects each having a small effect. Negative allelic effects had larger effects (Figure 8c) than the positive effects (Figure 8d). If implemented, the recommendation from this article for eliminating heifers with sharply negative homozygous recessive genotypes should help reduce the frequency of the large negative fertility effects in the U.S. Holstein population, although those negative genotypes already had low frequencies.

### 2.11. Comparison between the 2023 and 2019 Studies

The large-scale GWAS [8] (2019 study) and the million-cow GWAS of this article (2023 study) were most comparable because these studies all used U.S. Holstein cows, the unprecedently large samples at the time of each study, and the same methods of data analysis. Therefore, the comparison with previous studies was focused on the comparison between the 2023 and 2019 studies. However, these two studies had major differences.

A major difference between these two studies was the different sample sizes, with the sample sizes of the 2023 study being 4.28–5.38-times as large as those of the 2019 study (Table 6). Two other major differences were the different versions of the cattle genome assembly and the different numbers of SNPs for the GWAS. The 2019 study used the UMD 3.1 version of the cattle genome assembly and 60,671 SNPs (60K), whereas the 2023 study used the ARS-UCD1.3 version of the cattle genome assembly and 78,964 SNPs (80K). The 80K and 60K had 44,718 common SNPs; the 80K had 34,246 SNPs not in the 60K; the 60K had 15,953 SNPs not in the 80K. Therefore, the estimates of the significant effects of the 60K confirmed by the 80K of the 2023 study should only be interpreted as approximate estimates of the confirmed effects by the 2023 study.

All additive effects of the 2019 study were compared with the top-1000 additive effects for DPR and CCR and with all 726 additive effects for HCR of the 2023 study. The results showed that 13–42% of additive effects from the 2019 study were confirmed by the 2023 study, with DPR having the highest percentage of confirmed effects and HCR having the lowest percentage of confirmed effects (Table 6). Given the sample sizes of the 2019 study with 269,158 cows for HCR, only 13% (2 out of 15) of the previous effects making the top 726 effects in the 2023 study were an indication of the challenging task of identifying genetic effects for a low-heritability trait such as HCR and was an indication that large samples such as the million-cow samples of the 2023 study could be extremely valuable for identifying the genetic effects of low-heritability traits. The results of DPR with 42% confirmed effects and CCR with 18% confirmed effects pointed to the same conclusions, which were further supported by the dominance results.

The 2019 study detected only five dominance effects for the three fertility traits involving two SNPs, one SNP in *AFF1* of Chr06 for all three fertility traits and one SNP in *SIPA1L3* of Chr18 for DPR and CCR. The dominance effects of the *AFF1* SNP were confirmed for all three fertility traits, whereas the dominance effects of the *SIPA1L3* SNP were not confirmed by the 2023 study, although the additive effect of the *SIPA1L3* SNP was among the top-20 additive effects for DPR in the 2023 study (Table 1). The 2023 study detected 69 more dominance effects than in the 2019 study (74 vs. 5). 

Some new effects of the 80K were due to new SNPs not in the 60K, and some effects in the 60K only were due to SNPs not in the 80K. The effects of SNPs in the 80K only included those in the *CEBPG-PEPD* and *DEDD2-POU2F2* region of Chr18 for DPR and CCR and in *KLHL8* and *PAG6* for HCR. The most-notable 60K-only SNP effect was that in *COX17*, which had the second-most-significant effect for DPR and the most-significant effect for CCR. The most-notable effects confirmed by both the 2023 and 2019 studies included those in the *SLC4A4-GC*-*NPFFR2* region of Chr06 and in *KALRN* of Chr01 for DPR and CCR and in *AFF1* of Chr06 for all three fertility traits.

The comparison between HCR results of this study and the AFC results in a previous study [9] led to an interesting inference about the potential differences of the genetic effects with respect to the conception rate and successful pregnancy. The HCR effects (Appendix A) overlapped many of the AFC effects on Chr19, but had no overlap with the AFC effects on Chr15, which along with Chr19 had most of the significant AFC effects. Since AFC involves puberty, conception, and successful pregnancy till birth, the Chr19 AFC effects should be contributing to the conception component of AFC, whereas the Chr15 AFC effects should be contributing to the successful pregnancy given that the Chr15 AFC effects had no overlap with the HCR effects. It is interesting that the progesterone receptor (*PGR*) gene of Chr15 known to be associated with the maintenance of pregnancy had a highly significant effect on AFC, but had no effect on HCR, consistent with the assumption that the Chr15 effects of AFC were related to the pregnancy maintenance of heifers.

This 2023 study detected some effects in or near genes known to affect reproduction that the 2019 study did not detect, including *SHBG* (Figure 1a), *GNRHR* (Figure 1c), *ESR1* (Figure 4e), and multiple genes of pregnancy-associated glycoproteins (Figure 5f). *SHBG* is the sex-hormone-binding globulin gene and is likely associated with early puberty [11,12]. *GNRHR* is gonadotropin-releasing hormone receptor, and the activation of the receptor ultimately causes the release of gonadotropic luteinizing hormone (LH) and follicle-stimulating hormone (FSH) [13]. *ESR1* is the estrogen receptor 1 gene, and the protein encoded by this gene regulates the transcription of many estrogen-inducible genes that play a role in growth, metabolism, sexual development, gestation, and other reproductive functions and is expressed in many non-reproductive tissues [14]. Pregnancy-associated glycoproteins are correlated with pregnancy success and are predictive of impending embryonic mortality in both beef and dairy cattle [15]. These known reproductive functions should increase the likelihood that genetic effects in or near these genes were true effects. 

### 2.12. Gene Ontology of Candidate Genes

To understand the potential biological functions of the candidate genes, we searched the Gene Ontology Resources (GO) [16] for the biological processes involved for the candidate genes of the additive and dominance effects for DPR (Table 1 and Table 2) and HCR (Table 3 and Table 4). A major question to answer for the GO search was whether candidate genes with highly significant effects such as *GC*, *NPFFR2*, and *SLC4A4* had known reproductive functions, but the GO results found none for DPR (Appendix A; Appendix A), noting that the candidate genes with known reproductive functions were not included in the GO analysis, such as *SHBG* (Figure 1a), *GNRHR* (Figure 1c), *ESR1* (Figure 4e), and multiple genes of pregnancy-associated glycoproteins (Figure 5f). The GO results for the HCR additive effects found one candidate gene (*MYOCD*) involving reproduction (development of the reproductive system and structure; Appendix A, Appendix A). No candidate gene of the dominance effects was found to be involved in reproduction. The GO results of DPR and HCR showed that every candidate gene involved multiple biological processes, and some biological processes involved multiple genes (Appendix A). *GC* (GC vitamin-D-binding protein) was involved in vitamin D and steroid metabolic processes, and *NPFFR2* (neuropeptide FF receptor 2) and SLC4A4 (solute carrier family 4 member 4) each involved many regulatory processes (Appendix A). However, none of those regulatory processes was known to affect reproduction. In contrast to the lack of evidence for the connections between the known biological functions of the candidate genes and the phenotypes, the GWAS results in this study provided Holstein-specific and high-confidence evidence for the association between many candidate genes and the three fertility traits. The combination of the GO results with the GWAS results of this study should point to the potential involvement of various regulatory processes of *NPFFR2* and *SLC4A4*, as well as the vitamin D and steroid metabolism of *GC* in Holstein female fertility. Similarly, the combination of the GO and GWAS results should provide useful indications of the potential genetic mechanisms of other significant SNP effects affecting the fertility traits in Holstein cows.

## 3. Materials and Methods

### 3.1. Holstein Population and SNP Data

The Holstein population in this study had 1,001,374–1,194,736 first-lactation Holstein cows with phenotypic observations on the daughter pregnancy rate (DPR), cow conception rate (CCR), and heifer conception rate (HCR) and genotypes of 78,964 original and imputed SNPs. The SNP genotypes were from 32 SNP chips with various densities (Appendix A) and were imputed to 78,964 SNPs via the FindHap algorithm [17] as a routine procedure for genomic evaluation by the Council on Dairy Cattle Breeding (CDCB) [18]. The SNP genotyping quality control by the CDCB had checks and requirements at the individual and SNP levels, including the call rate, parent–progeny conflicts, sex verification using X-specific SNPs, and Hardy–Weinberg equilibrium [19,20]. In addition, we applied a minor allele frequency (MAF) of 5% for SNP filtering in this study. The phenotypic values used in the GWAS analysis were the phenotypic residuals after removing fixed non-genetic effects available from the December 2022 U.S. Holstein genomic evaluation by the CDCB. The basic statistics of the cows and the phenotypic data of the three fertility traits are given in Appendix A. With the requirement of a 5% MAF, the number of SNPs for the GWAS analysis was 75,524 for DPR, 75,295 for CCR and 75,140 for HCR. For 75,524 SNPs with additive and dominance effects, the threshold *p*-value for declaring significant *t*-tests for the Bonferroni correction with 0.05 genome-wide false positives was 10^−8^, or log_10_(1/p) = 8. The SNP and gene positions were those from the ARS-UCD1.3 cattle genome assembly [21]. Genes containing or in the proximity of highly significant additive and dominance effects were identified as candidate genes affecting the three fertility traits. All figures for the GWAS results were produced using SNPEVG2 in the SNPEVG package [22]. 

### 3.2. GWAS Analysis 

The GWAS analysis used an approximate generalized least-squares (AGLS) method. The AGLS method combines the least-squares (LS) tests implemented by EPISNP1mpi [23,24] with the estimated breeding values from routine genetic evaluation using the entire U.S. Holstein population. The statistical model was:(1)$$y=\mu I+X_{g}g+Za+e=Xb+Za+e$$
where **y** = column vector of phenotypic deviation after removing fixed nongenetic effects such as heard–year–season (termed as “yield deviation” for any trait) using a standard procedure for the CDCB/USDA genetic and genomic evaluation; µ = common mean; **I** = identity matrix; **g** = column vector of genotypic values; $X_{g}$ = model matrix of **g**; b=(μ, g′)′, $X=(I,\;X_{g})$; **a** = column vector of additive polygenic values; **Z** = model matrix of **a**; and **e** = column vector of random residuals. The first and second moments of Equation (1) are:E(y)=Xb and $\mathrm{var}(y)=V=ZGZ\prime +R=\sigma_{a}^{2}ZAZ\prime +\sigma_{e}^{2}I$, where $\sigma_{a}^{2}$ = additive variance, **A** = additive relationship matrix, and $\sigma_{e}^{2}$ = residual variance. The problem of estimating the **b** vector that includes SNP genotypic values in Equation (1) is the requirement of inverting the **V** if the generalized least-squares (GLS) method is used or inverting the **A** matrix if the mixed model equations (MMEs) [25] are used. However, both **V** and **A** cannot be inverted for our sample size. To avoid inverting these large matrices, the GWAS used the method of approximate GLS (AGLS), which replaces the polygenic additive values (**a**) with the best linear unbiased prediction based on pedigree relationships [8]. The AGLS method is based on the following results:(2)$$\hat{b}={\left(X\prime V^{-1}X\right)}^{-}X\prime V^{-1}y$$
(3)$$\begin{array}{ll} \hat{b} & =(X\prime R^{-1}X{)}^{-}(X\prime R^{-1}y-X\prime R^{-1}Z\hat{a})\\ \; & =(X\prime X{)}^{-}X\prime (y-Z\hat{a})=(X\prime X{)}^{-}X\prime y_{*} \end{array}$$
where $y_{*}=y-Z\hat{a}$ and $\hat{a}$ is the best linear unbiased prediction (BLUP) of **a**. Equation (2) is the GLS solution, and Equation (3) is the MME solution of **b**. These two equations yield identical results, and $\hat{b}$ from either equation is termed the best linear unbiased estimator (BLUE) [25]. If $\hat{a}$ is known, the LS version of the BLUE given by Equation (3) is computationally efficient relative to the GLS of Equation (2), requiring the **V** inverse, or the joint MME solutions of $\hat{b}$ and $\hat{a}$, requiring the **A** inverse. The AGLS method uses two approximations. The first approximation is to use $\tilde{a}$ from routine genetic evaluation as an approximation of $\hat{a}$ in Equation (3):(4)$$\hat{b}=(X\prime X{)}^{-}X\prime (y-Z\tilde{a})=(X\prime X{)}^{-}X\prime y^{*}$$
where $y^{*}=y-Z\tilde{a}$, and $\tilde{a}$ is the column vector of 2(PTA) with PTA being the predicted transmission ability from the routine genetic evaluation. Equation (4) achieves the benefit of sample stratification correction from mixed models using pedigree relationships without the computing difficulty of inverting **V** or **A**. The second approximation of the AGLS approach is the t-test using the LS, rather than the GLS formula of the t-statistic, to avoid using the **V** inverse in the GLS formula. The significance tests for additive and dominance SNP effects used the t-tests of the additive and dominance contrasts of the estimated SNP genotypic values [23,26]. The t-statistic of the AGLS was calculated as:(5)$$t_{j}=\frac{{|L}_{j}|}{\sqrt{{\mathrm{var}(L}_{j})}}=\frac{\left|s_{j}^{}\hat{g}\right|}{v\sqrt{s_{j}^{}{\left(X_{}^{\prime }X_{}\right)}_{\mathrm{gg}}^{-}s_{j}^{\prime }}},\;j=a,d$$
where $L_{j}$ = additive or dominance contrast; $\sqrt{{\mathrm{var}(L}_{j})}$ = standard deviation of the additive or dominance contrast; $s_{a}$ = row vector of additive contrast coefficients = $\left[\begin{array}{ccc} P_{11}{/p}_{1} & 0{.5P}_{12}{(p}_{2}-p_{1})/(p_{1}p_{2}) & -P_{22}{/p}_{2} \end{array}\right]$; $s_{d}$ = row vector of dominance contrast coefficients = $\left[\begin{array}{ccc} -0.5 & 1 & 0.5 \end{array}\right]$; $v^{2}=\left(y-X\hat{b}{)}^{\prime }(y-X\hat{b}\right)/(n-k)$= estimated residual variance; $\hat{g}$ = column vector of the AGLS estimates of the three SNP genotypic effects of $g_{11}$, $g_{12}$, and $g_{22}$ from Equation (4); ${\left(X_{}^{\prime }X_{}\right)}_{\mathrm{gg}}^{-}$ = submatrix of ${\left(X_{}^{\prime }X\right)}_{}^{-}$ corresponding to $\hat{g}$; where $p_{1}$ = frequency of $A_{1}$ allele, $p_{2}$ = frequency of $A_{2}$ allele of the SNP, $P_{11}$ = frequency of $A_{1}A_{1}$ genotype, $P_{12}$ = frequency of $A_{1}A_{2}$ genotype, $P_{22}$ = frequency of $A_{2}A_{2}$ genotype, n = number of observations, and k = rank of **X**. The formula of $s_{a}$ defined above allows Hardy–Weinberg disequilibrium [26] and simplifies to $\left[\begin{array}{ccc} p_{1} & p_{2}-p_{1} & -p_{2} \end{array}\right]$ under Hardy–Weinberg equilibrium. 

The additive effects of each SNP were estimated using three measures, the average effect of gene substitution, the allelic mean, and the allelic effect of each allele based on quantitative genetics definitions [26,27]. The allelic mean ($\mu_{i}$), the population mean of all genotypic values of the SNP (μ), the allelic effect ($a_{i}$), and the average effect of gene substitution of the SNP (α) are:(6)$$\mu_{1}=P_{11.1}g_{11}+\;0{.5P}_{12.1}g_{12}$$
(7)$$\mu_{2}=0{.5P}_{12.2}g_{12}+\;P_{22.2}g_{22}$$
(8)$$\mu =\sum_{i=1}^{2}p_{i}\mu_{i}$$
(9)$$a_{i}\;=\;\mu_{i}-\mu ,\;i=1,\;2$$
(10)$$\alpha \;=\;L_{a}=s_{a}^{}\hat{g}\;=\;a_{1}-a_{2}=\;\mu_{1}-\mu_{2}$$
where $P_{11.1}=P_{11}{/p}_{1}$, $P_{12.1}=P_{12}{/p}_{1}$, $P_{12.2}=P_{12}{/p}_{2}$, and $P_{22.2}=P_{22}{/p}_{2}$. The additive effect measured by the average effect of gene substitution of Equation (10) is the distance between the two allelic means or effects of the same SNP and is the fundamental measure for detecting SNP additive effects, as shown by the t-statistic of Equation (5). The allelic effects defined by Equation (9) provide an understanding of the effect size and direction of each allele. However, the allelic effect of Equation (9) is not comparable across SNPs because the allelic effect is affected by the genotypic mean of the SNP defined by Equation (8). To compare allelic effects across SNPs, we replaced the SNP genotypic mean (μ) in Equation (9) with the average of all SNP genotypic means ($\mu_{\mathrm{all}}$):(11)$$a_{i}\;=\;\mu_{i}-\mu_{\mathrm{all}},\;i=1,\;2$$

Equation (11) was used only for the purpose of graphical displaying the allelic effects. The dominance effect of each SNP was estimated as the dominance contrast of $\hat{g}$ from Equation (4):(12)$$\delta =L_{d}\;=\;d_{12}-{(d}_{11}{+d}_{22})/2\;=\;g_{12}-{(g}_{11}+g_{22})/2$$
where $g_{\mathrm{ij}}$ is the AGLS estimates of the SNP genotypic value from Equation (4) (i, j = 1, 2) and $d_{\mathrm{ij}}$ is the dominance value (dominance deviation) of the $A_{i}A_{j}$ SNP genotype:(13)$$d_{\mathrm{ij}}=g_{\mathrm{ij}}\;-\mu -a_{i}-a_{j}$$

In this study, overdominance refers to the fact that the dominance value of the heterozygous genotype is more extreme than that of either homozygous genotype, i.e., $d_{12}$ > $d_{11}$ and $d_{12}$ > $d_{22}$ for positive overdominance effects or $d_{12}$ < $d_{11}$ and $d_{12}$ < $d_{22}$ for negative overdominance effect. The genotypic means of the corrected phenotypic values after removing the pedigree additive (breeding) values from the original phenotypic values (Equations (1) and (4)), and the genotypic means of the original phenotypic values without removing the pedigree additive values were also used for the interpretation of the dominance effects. For the example of using the original phenotypic values, complete dominance of the dominant allele over the recessive allele was defined by $y_{\mathrm{rd}}{=y}_{\mathrm{dd}}$ (not observed in this study), nearly complete dominance by $y_{\mathrm{rd}}\;\approx {\;y}_{\mathrm{dd}}$ (observed in this study), partial dominance by $y_{\mathrm{rd}}{\;<\;}_{\mathrm{dd}}$ (observed in this study), and positive overdominance by $y_{\mathrm{rd}}{\;>\;y}_{\mathrm{dd}}$ and $y_{\mathrm{rd}}{\;>\;y}_{\mathrm{rr}}$ (not observed in this study), where $y_{\mathrm{rr}}$ = the mean of the original phenotypic values without removing the pedigree additive (breeding) values of cows with the homozygous recessive genotype, $y_{\mathrm{rd}}$ = the mean of the original phenotypic values without removing the pedigree additive (breeding) values of cows with the heterozygous genotype, and $y_{\mathrm{dd}}$ = the mean of the original phenotypic values without removing the pedigree additive (breeding) values of cows with the homozygous dominant genotype of the SNP. A neutral dominant allele was defined as the lack of deviation of the genotypic mean of the homozygous dominant genotype from the population mean and was considered to have no effect on the phenotype by the dominant allele when in homozygous status.

To evaluate the impact of sharply negative homozygous recessive genotypes, a measure of negative impact was calculated as the difference between the mean phenotypic values of cows with the homozygous recessive genotypes and the mean values of the other two genotypes, the homozygous dominant genotype and the heterozygous genotype: (14)$$\mathrm{NI}\;=\;y_{\mathrm{rr}}-{(y}_{\mathrm{rd}}{+y}_{\mathrm{dd}})/2$$
where NI = negative impact of the homozygous recessive genotype.

To understand the potential reproductive functions of selected candidate genes, Gene Ontology (GO) analysis was performed using the OmicShare platform (www.omicshare.com/tools, accessed on 12 June 2023).

## 4. Conclusions

The million-cow GWAS identified many additive effects and a small number of dominance effects affecting the three fertility traits (DPR, CCR, HCR) in U.S. Holstein cows with high statistical confidence. The additive effects mostly were small effects, whereas the dominance effects mostly were substantially larger than the additive effects. Each dominance effect typically involved a dominant allele and a recessive allele, where the recessive allele had a sharply negative effect when in homozygous status and the dominant allele neutralized most of the negative effect of the recessive allele when in heterozygous status, but behaved like a neutral allele when in homozygous status. Most of the effects were new; some effects confirmed previously reported effects; some effects involved candidate genes with known functions affecting reproduction. The results from this study provided a new understanding of genetic factors underlying the three fertility traits in U.S. Holstein cows. 

## Figures and Tables

**Figure 1 ijms-24-10496-f001:**
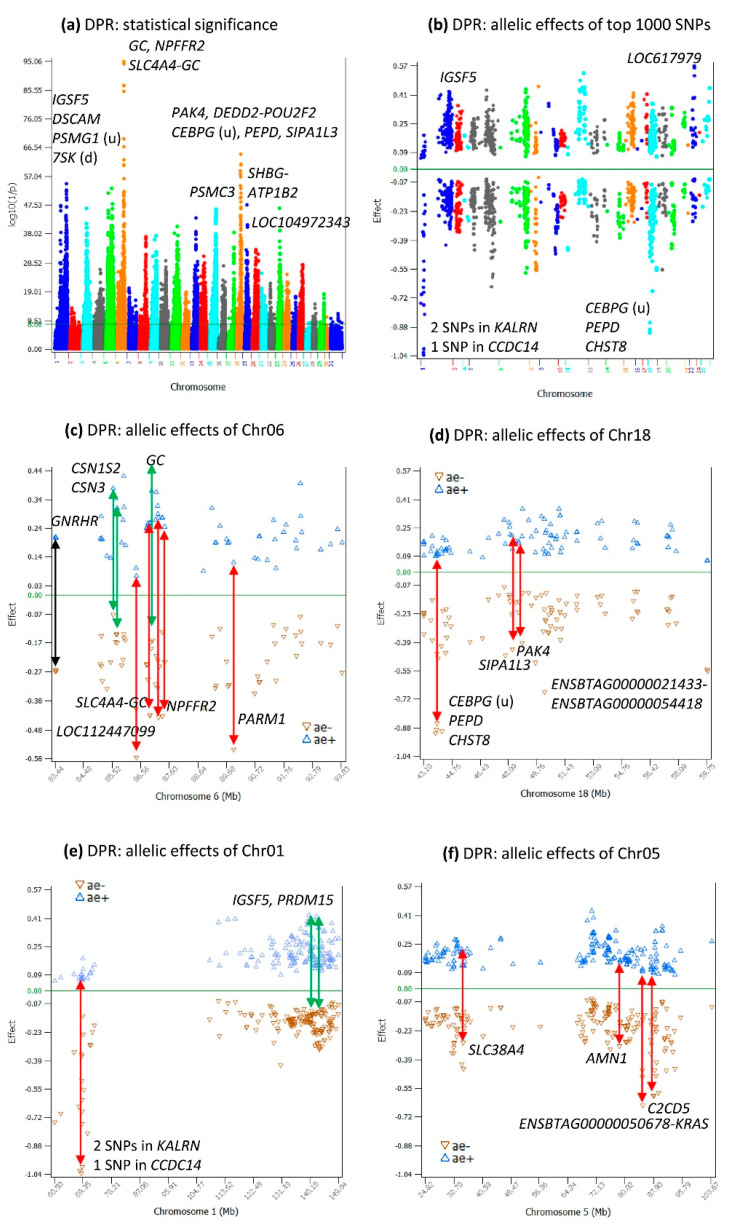
Additive effects of DPR. (**a**) Manhattan plot of genome-wide additive effects; (**b**) positive and negative allelic effects of the top-1000 most-significant additive effects; (**c**) allelic effects of Chr06 among the top-1000 additive effects; (**d**) allelic effects of Chr18 among the top-1000 additive effects; (**e**) allelic effects of Chr01 among the top-1000 additive effects; (**f**) allelic effects of Chr05 among the top-1000 additive effects. The black double-arrowed line indicates symmetric allelic effects. The green double-arrowed line indicates a larger effect size of the positive allele than the negative allele. The red double-arrowed line indicates a larger effect size of the negative allele than the positive allele. Chr30 is the pseudoautosome region of the X chromosome. Chr31 is the nonrecombining region of the X chromosome (no recombination with the Y chromosome).

**Figure 2 ijms-24-10496-f002:**
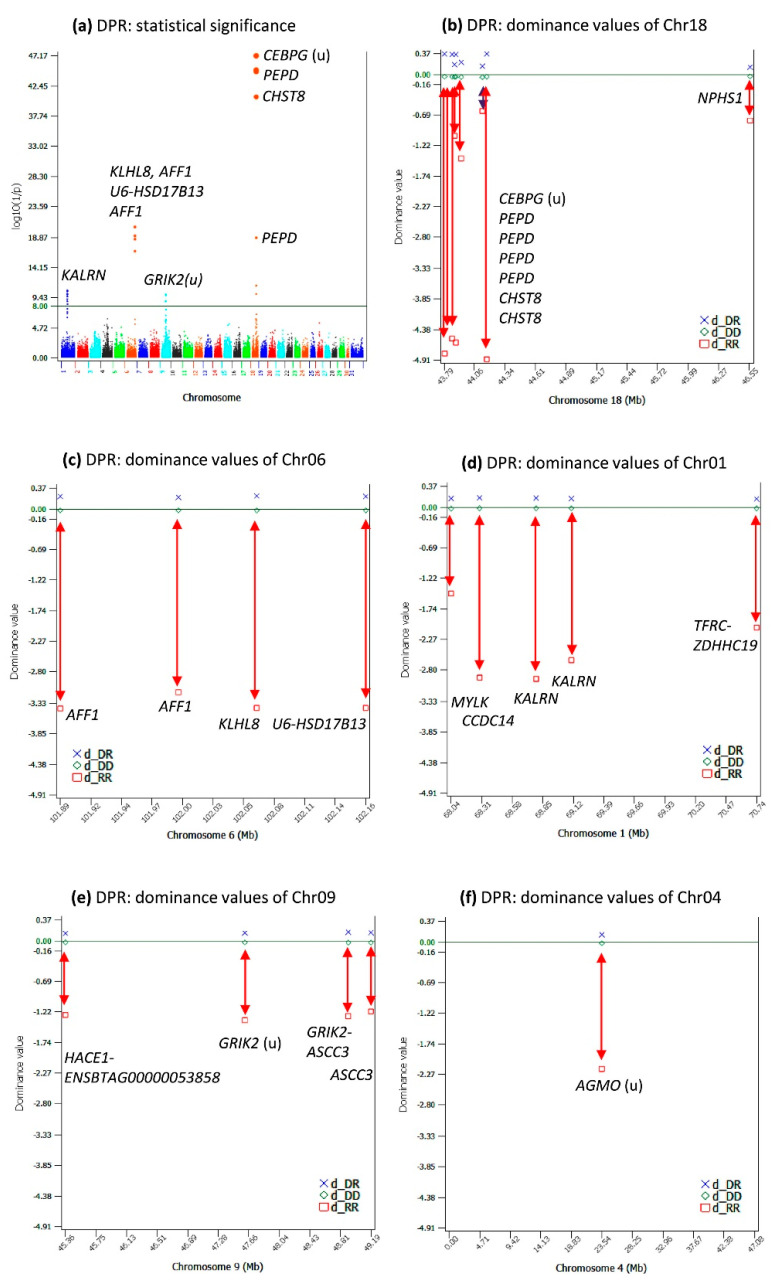
Dominance effects of DPR. (**a**) Manhattan plot of the statistical significance of genome-wide dominance effects; (**b**) dominance values of Chr18; (**c**) dominance values of Chr06; (**d**) dominance values of Chr01; (**e**) dominance values of Chr09; (**f**) dominance values of Chr04. The red double-arrowed line indicates the negative dominance value of the homozygous recessive genotype. Chr30 is the pseudoautosome region of the X chromosome. Chr31 is the nonrecombining region of the X chromosome (no recombination with the Y chromosome).

**Figure 3 ijms-24-10496-f003:**
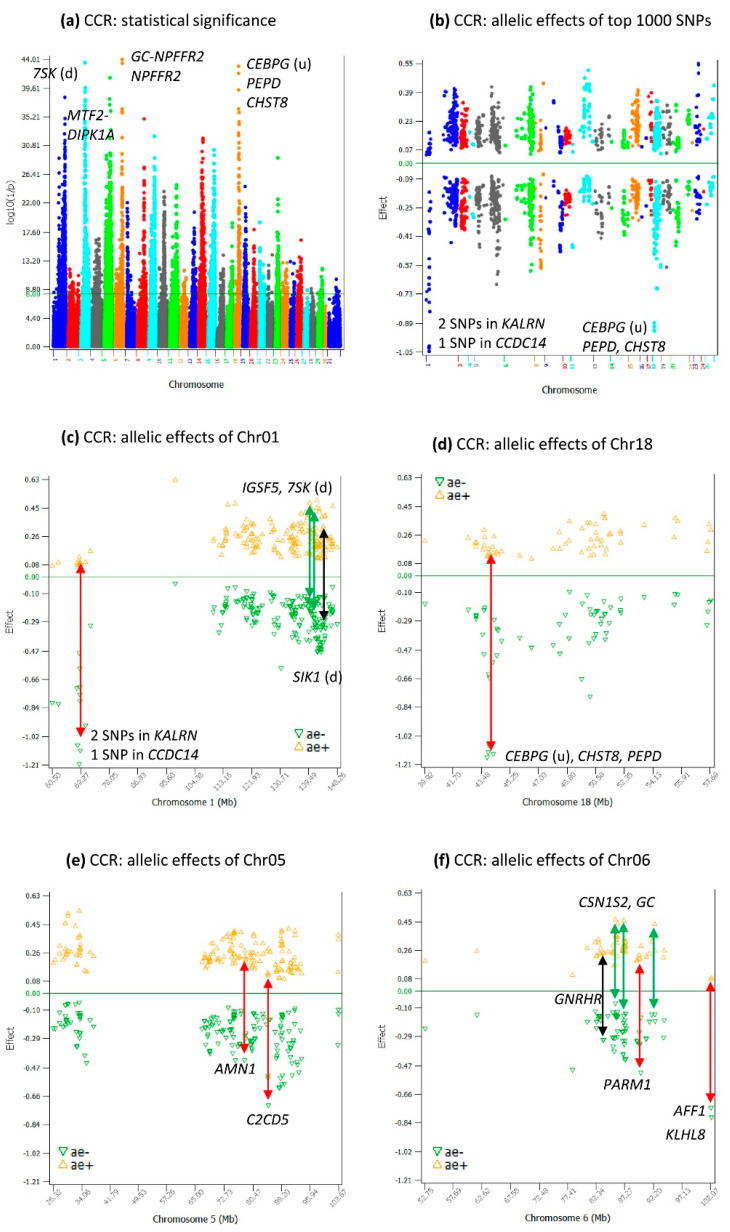
Additive effects of CCR. (**a**) Manhattan plot of the genome-wide additive effects; (**b**) positive and negative allelic effects of the top-1000 most-significant additive effects; (**c**) allelic effects of Chr01 among the top-1000 additive effects; (**d**) allelic effects of Chr18 among the top-1000 additive effects; (**e**) allelic effects of Chr05 among the top-1000 additive effects; (**f**) allelic effects of Chr06 among the top-1000 additive effects. The black double-arrowed line indicates symmetric allelic effects. The green double-arrowed line indicates a larger effect size of the positive allele than the negative allele. The red double-arrowed line indicates a larger effect size of the negative allele than the positive allele. Chr30 is the pseudoautosome region of the X chromosome. Chr31 is the nonrecombining region of the X chromosome (no recombination with the Y chromosome).

**Figure 4 ijms-24-10496-f004:**
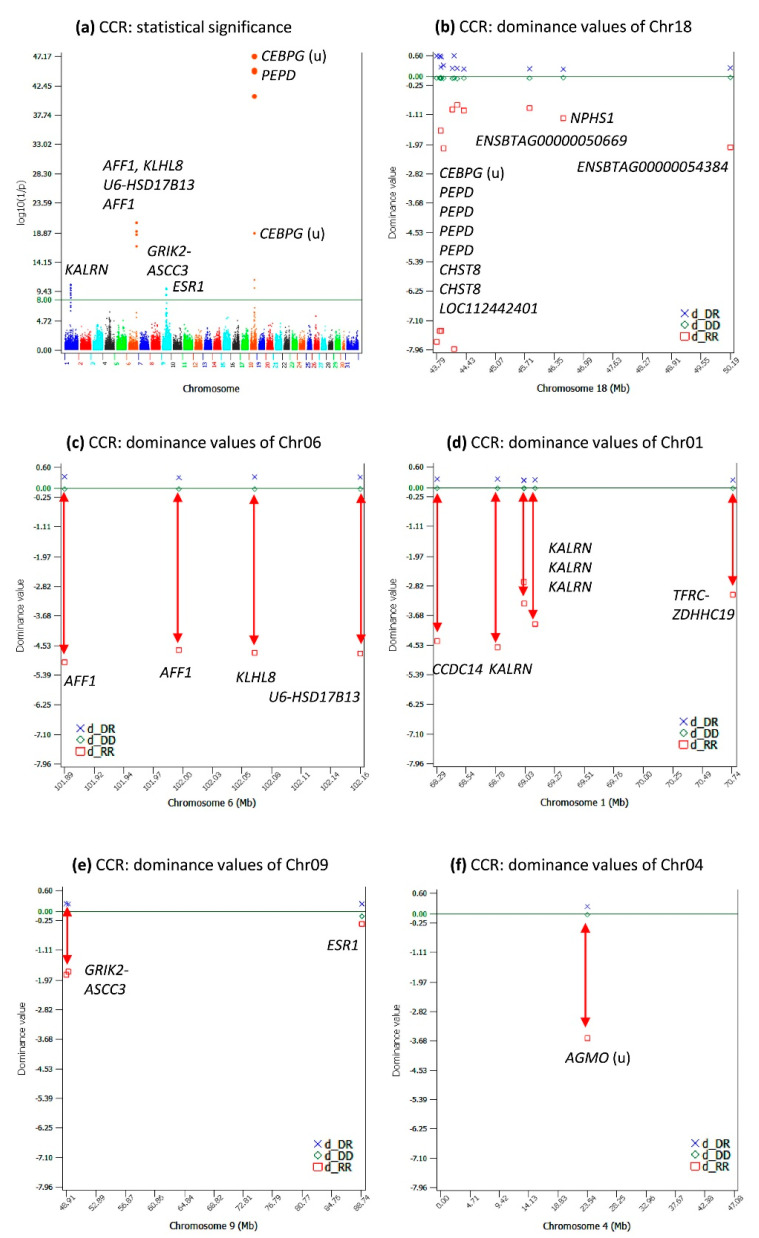
Dominance effects of DPR. (**a**) Manhattan plot of the statistical significance of genome-wide dominance effects; (**b**) dominance values of Chr18; (**c**) dominance values of Chr06; (**d**) dominance values of Chr01; (**e**) dominance values of Chr09; (**f**) dominance values of Chr04. The red double-arrowed line indicates the negative dominance value of the homozygous recessive genotype. Chr30 is the pseudoautosome region of the X chromosome. Chr31 is the nonrecombining region of the X chromosome (no recombination with the Y chromosome).

**Figure 5 ijms-24-10496-f005:**
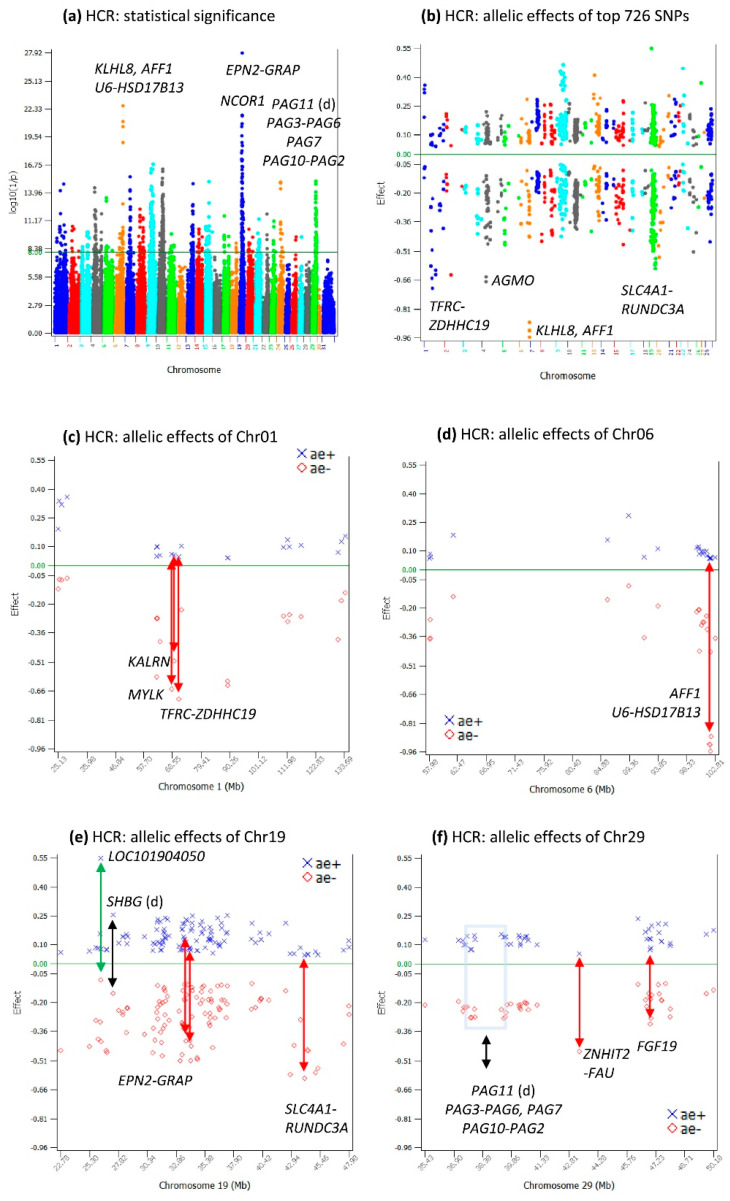
Additive effects of HCR. (**a**) Manhattan plot of the genome-wide additive effects; (**b**) positive and negative allelic effects of the top-1000 most-significant additive effects; (**c**) allelic effects of Chr01 among the top-1000 additive effects; (**d**) allelic effects of Chr06 among the top-1000 additive effects; (**e**) allelic effects of Chr19 among the top-1000 additive effects; (**f**) allelic effects of Chr29 among the top-1000 additive effects. The black double-arrowed line indicates symmetric allelic effects. The green double-arrowed line indicates a larger effect size of the positive allele than the negative allele. The red double-arrowed line indicates a larger effect size of the negative allele than the positive allele. Chr30 is the pseudoautosome region of the X chromosome. Chr31 is the nonrecombining region of the X chromosome (no recombination with the Y chromosome).

**Figure 6 ijms-24-10496-f006:**
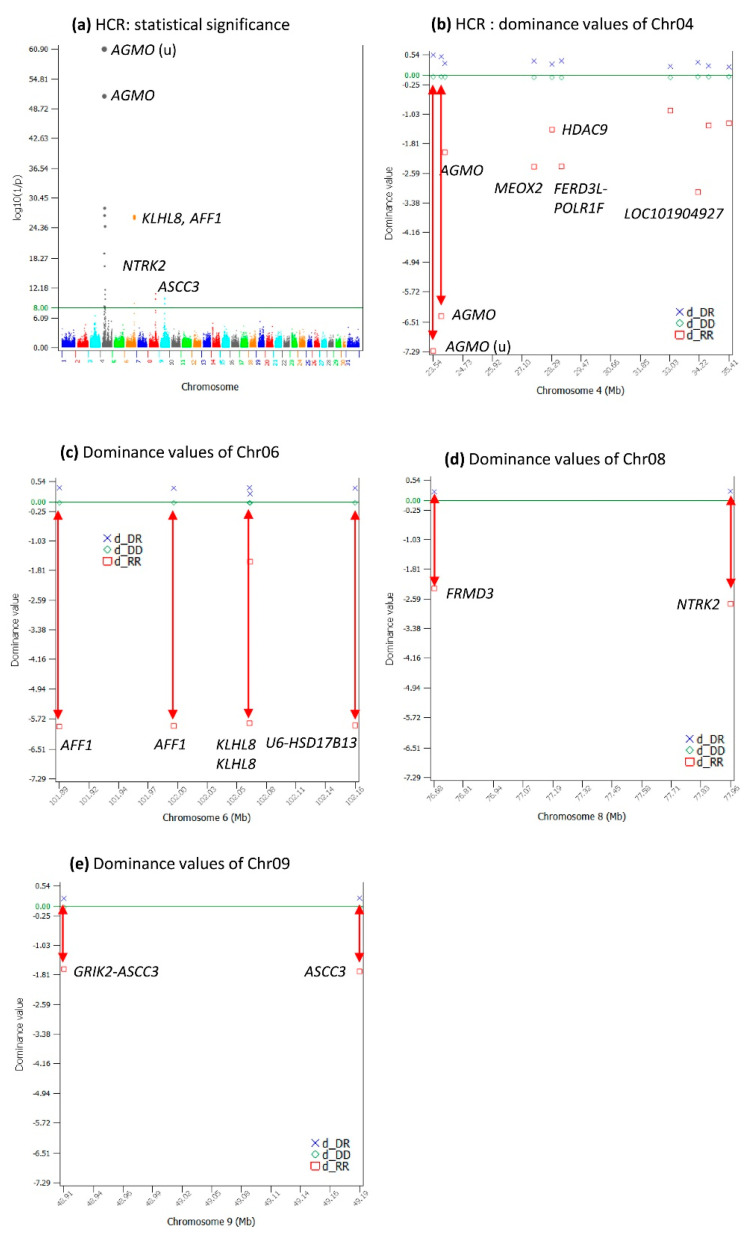
Dominance effects of DPR. (**a**) Manhattan plot of the statistical significance of genome-wide dominance effects; (**b**) dominance values of Chr04; (**c**) dominance values of Chr06; (**d**) dominance values of Chr08; (**e**) dominance values of Chr09. The red double-arrowed line indicates the negative dominance value of the homozygous recessive genotype. Chr30 is the pseudoautosome region of the X chromosome. Chr31 is the nonrecombining region of the X chromosome (no recombination with the Y chromosome).

**Figure 7 ijms-24-10496-f007:**
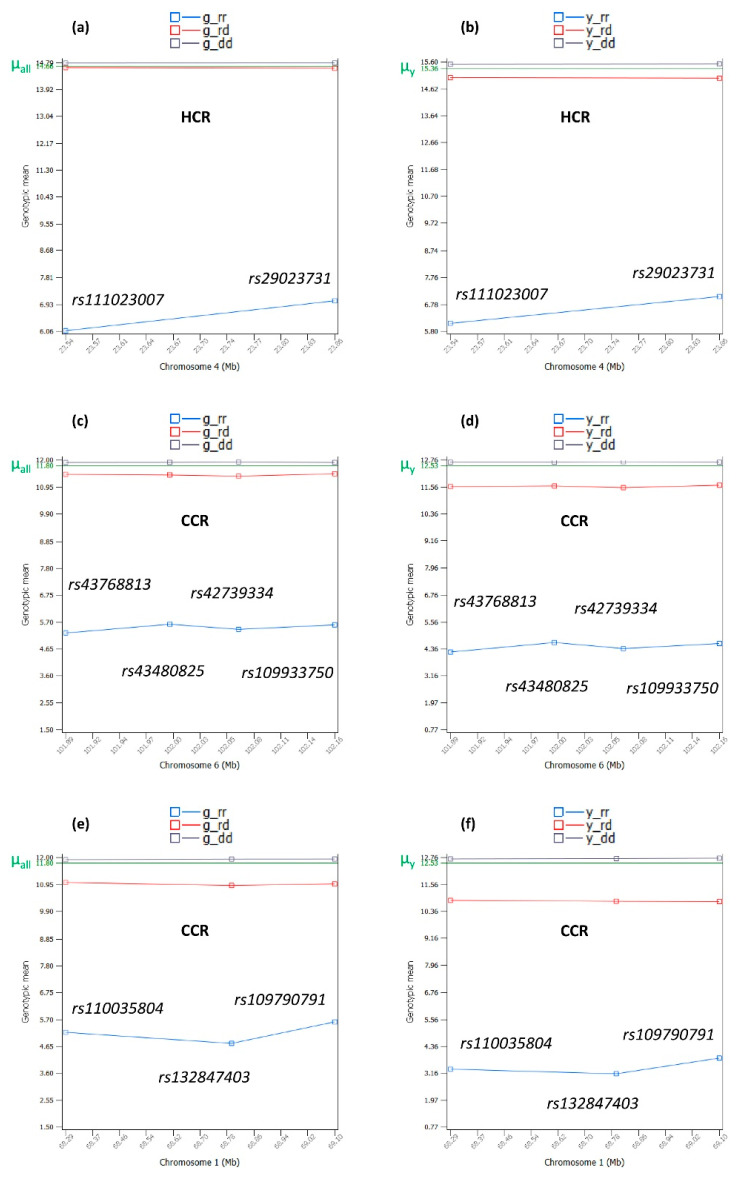
Examples of the SNP genotypic means with sharply negative dominance values for the three fertility traits. (**a**,**b**) Genotypic means of the phenotypic values showing a nearly complete dominance of the dominant allele over the recessive allele. (**c**,**d**) Genotypic means of the phenotypic values showing a less-complete dominance of the dominant allele over the recessive allele than in (**a**,**b**). (**e**,**f**) Genotypic means of the phenotypic values showing partial dominance of the dominant allele over the recessive allele. Each dominant allele was a neutral allele when in homozygous status. g_ij = genotypic mean of the corrected phenotypic values after removing the pedigree additive (breeding) values, and y_ij = genotypic mean of the original phenotypic values without removing the pedigree additive (breeding) values, where i or j = r indicates the recessive allele and i or j = d indicates the dominant allele. µ_all_ is the average of the genotypic means of all SNPs. µ_y_ is the average of the phenotypic values of all cows.

**Figure 8 ijms-24-10496-f008:**
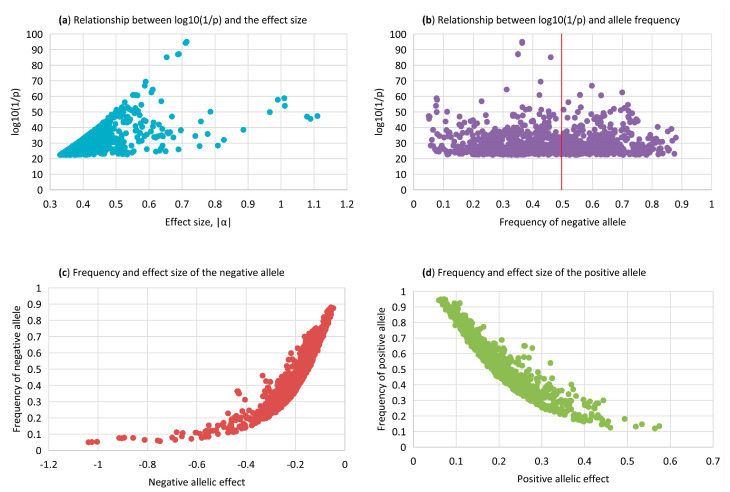
Relationships among statistical significance, effect size, and allele frequencies for daughter pregnancy rate (DPR). (**a**) Relationship between statistical significance measured by the log_10_(1/p) and the size of the additive effect; (**b**) relationship between statistical significance measured by the log_10_(1/p) and the frequency of the negative allele; (**c**) frequency of each allele with negative allelic effect; (**d**) frequency of each allele with positive allelic effect.

**Table 1 ijms-24-10496-t001:** Top-20 additive effects of the daughter pregnancy rate (DPR).

SNP	Chr	Position	Candidate Gene	α	al+	ae+	f_al+	al−	ae−	f_al−	log_10_(1/p)
*rs110434046*	6	87184768	*GC-NPFFR2*	−0.714	2	0.260	0.636	1	−0.454	0.364	95.06
*rs109034709*	6	87316810	*NPFFR2*	−0.711	2	0.258	0.636	1	−0.452	0.364	94.27
*rs110380398*	6	86877334	*SLC4A4-GC*	0.690	1	0.242	0.650	2	−0.448	0.350	87.16
*rs110527224*	6	86860291	*SLC4A4-GC*	−0.687	2	0.241	0.650	1	−0.447	0.350	87.00
*rs137147462*	6	87153414	*GC-NPFFR2*	−0.653	2	0.301	0.540	1	−0.353	0.460	85.12
*rs109452259*	6	87068809	*GC-NPFFR2*	0.590	1	0.251	0.574	2	−0.338	0.426	69.45
*rs42767507*	6	87101644	*GC-NPFFR2*	−0.587	2	0.351	0.402	1	−0.236	0.598	66.81
*rs210926829*	18	48869465	*PAK4*	0.611	1	0.191	0.688	2	−0.42	0.312	64.40
*rs134342291*	6	86964714	*GC*	0.607	1	0.425	0.300	2	−0.182	0.700	62.52
*rs41635776*	18	51084210	*DEDD2-POU2F2*	−0.552	2	0.308	0.442	1	−0.244	0.558	61.02
*rs137664040*	6	86795926	*SLC4A4*	−0.560	2	0.236	0.578	1	−0.324	0.422	60.86
*rs110682631*	18	51107045	*DEDD2-POU2F2*	−0.551	2	0.307	0.442	1	−0.244	0.558	60.82
*rs43338539*	6	86924174	*SLC4A4-GC*	0.565	1	0.355	0.371	2	−0.21	0.629	60.63
*rs41884737*	18	43786051	*CEBPG* (u)	1.010	1	0.077	0.923	2	−0.933	0.077	58.80
*rs41885943*	18	43854199	*PEPD*	0.990	1	0.078	0.922	2	−0.912	0.078	57.79
*rs109972395*	18	47886323	*SIPA1L3*	−0.637	2	0.145	0.772	1	−0.492	0.228	56.86
*rs42766480*	6	87156735	*GC-NPFFR2*	0.526	1	0.273	0.481	2	−0.253	0.519	56.23
*rs43275844*	1	139828817	*IGSF5*	−0.576	2	0.414	0.281	1	−0.162	0.719	54.59
*rs43746558*	18	44167440	*CHST8*	1.012	1	0.076	0.925	2	−0.936	0.075	53.94
*rs109118753*	5	78228782	*AMN1*	−0.534	2	0.195	0.635	1	−0.339	0.365	53.02

“u” indicates that the SNP is upstream of the gene. “α” is the additive effect of the SNP as the difference between the allelic effects of “allele 1” and “allele 2” (Equation (10)). “ae+” is the allelic effect of the positive allele (Equation (11)). “ae−“ is the allelic effect of the negative allele (Equation (11)). “f_al+” is the frequency of the positive allele. “f_al−“ is the frequency of the negative allele.

**Table 2 ijms-24-10496-t002:** Dominance effects for the daughter pregnancy rate (DPR) greater than the largest additive effect (δ > 1.11).

SNP	Chr	Position	Candidate Gene	δ	DR	d_DR	f_DR	DD	d_DD	f_DD	RR	d_RR	f_RR	log_10_(1/p)
*rs41884737* ^a^	18	43786051	*CEBPG* (u)	2.78	12	0.365	0.142	11	−0.030	0.852	22	−4.81	0.005	59.30
*rs41885943* ^a^	18	43854199	*PEPD*	2.64	12	0.355	0.145	11	−0.030	0.849	22	−4.55	0.006	56.09
*rs43746558* ^a^	18	44167440	*CHST8*	2.84	12	0.366	0.139	11	−0.029	0.856	22	−4.91	0.005	55.36
*rs133443778* ^a^	18	43887966	*PEPD*	2.68	12	0.354	0.143	11	−0.029	0.851	22	−4.62	0.006	51.30
*rs42739334*	6	102065812	*KLHL8*	1.95	12	0.235	0.123	11	−0.016	0.873	22	−3.42	0.004	23.13
*rs109933750*	6	102164971	*KLHL8-HSD17B13*	1.95	12	0.226	0.121	22	−0.016	0.875	11	−3.42	0.004	22.00
*rs43768813*	6	101887271	*AFF1*	1.95	12	0.227	0.119	22	−0.015	0.877	11	−3.43	0.004	21.76
*rs43480825*	6	101994654	*AFF1*	1.79	12	0.210	0.122	11	−0.015	0.874	22	−3.15	0.004	19.07
*rs110035804*	1	68293671	*CCDC14*	1.64	12	0.170	0.097	11	−0.009	0.900	22	−2.93	0.003	10.76
*rs132847403*	1	68792344	*KALRN*	1.64	12	0.167	0.095	11	−0.009	0.902	22	−2.95	0.003	10.59
*rs111023007*	4	23542497	*AGMO* (u)	1.23	12	0.137	0.139	11	−0.011	0.856	22	−2.18	0.004	9.94
*rs109790791*	1	69102381	*KALRN*	1.48	12	0.158	0.100	11	−0.009	0.897	22	−2.63	0.003	9.48
*rs137814900*	1	70735271	*TFRC-ZDHHC19*	1.19	12	0.154	0.121	11	−0.011	0.875	22	−2.07	0.004	9.02

^a^ This SNP is recommended for culling heifers with the homozygous recessive genotype based on its effect on DPR (δ > 2.0). “u” indicates the SNP is upstream of the gene. “δ” is the dominance effect of the SNP as the difference between the heterozygous dominance value and the average of the two homozygous dominance values (Equation (12)). “DR” is the heterozygous genotype with one dominant allele and one recessive allele. “d_DR” is the dominance value of the heterozygous genotype with one dominant allele (D) and one recessive allele (R) (Equation (13)). “DD” is the homozygous genotype with two dominant alleles. “d_DD” is the dominance value of the homozygous genotype with two dominant alleles (DD) (Equation (13)). “RR” is the homozygous genotype with two recessive alleles. “d_RR” is the dominance value of the homozygous genotype with two recessive alleles (RR) (Equation (13)). “f_DR” is the frequency of the heterozygous genotype. “f_DD” is the frequency of the homozygous genotype of the dominant allele. “f_RR” is the frequency of the homozygous genotype of the recessive allele.

**Table 3 ijms-24-10496-t003:** Top-20 additive effects of the heifer conception rate (HCR).

SNP	Chr	Position	Candidate Gene	α	al+	ae+	f_al+	al−	ae−	f_al−	log_10_(1/p)
*rs41911772*	19	34170903	*EPN2-GRAP*	0.606	1	0.174	0.714	2	−0.433	0.286	27.92
*rs42739334*	6	102065812	*KLHL8*	1.026	1	0.064	0.938	2	−0.962	0.062	22.65
*rs135712994*	19	33421057	*NCOR1*	−0.502	2	0.182	0.638	1	−0.320	0.362	21.71
*rs110569179*	19	34245953	*SLC5A10-FAM83G*	0.493	1	0.214	0.565	2	−0.279	0.435	21.63
*rs43480825*	6	101994654	*AFF1*	0.989	1	0.061	0.938	2	−0.928	0.062	21.07
*rs110761858*	19	33358794	*NCOR1*	−0.489	2	0.179	0.633	1	−0.310	0.367	20.85
*rs134340085*	19	28503389	*NTN1*	0.516	1	0.154	0.701	2	−0.362	0.299	20.56
*rs43768813*	6	101887271	*AFF1*	−0.985	2	0.060	0.939	1	−0.925	0.061	20.56
*rs109547103*	19	30101489	*ENSBTAG00000049618*	0.513	1	0.142	0.723	2	−0.371	0.277	19.65
*rs109933750*	6	102164971	*HSD17B13* (u)	−0.942	2	0.058	0.939	1	−0.884	0.061	18.97
*rs109086091*	19	31844296	*HS3ST3A1-COX10*	−0.483	2	0.159	0.671	1	−0.324	0.329	18.97
*rs29015945*	19	35808547	*SPAG9* (d)	0.452	1	0.191	0.577	2	−0.261	0.423	18.67
*rs135166134*	19	31231460	*ARHGAP44*	−0.450	2	0.220	0.510	1	−0.229	0.490	18.15
*rs41641007*	19	26224813	*RABEP1*	−0.463	2	0.157	0.661	1	−0.306	0.339	17.94
*rs3423456831*	19	26278782	*LOC101904050*	0.639	1	0.554	0.133	2	−0.085	0.867	17.30
*rs133654564*	19	31638013	*HS3ST3A1* (u)	−0.584	2	0.100	0.829	1	−0.484	0.171	17.29
*rs110940549*	19	33716989	*SPECC1*	0.520	1	0.113	0.782	2	−0.406	0.218	17.25
*rs3423444153*	19	31479267	*HS3ST3A1* (u)	0.434	1	0.221	0.491	2	−0.213	0.509	17.20
*rs3423447585*	19	31166010	*MYOCD*	−0.435	2	0.179	0.588	1	−0.256	0.412	17.15
*rs3423293217*	9	70653077	blank	−0.444	2	0.279	0.371	1	−0.165	0.629	16.80

“u” indicates the SNP is upstream of the gene, and “d” indicates the SNP is downstream of the gene. “α” is the additive effect of the SNP as the difference between allelic effects of “allele 1” and “allele 2” (Equation (10)). “ae+” is the allelic effect of the positive allele (Equation (11)). “ae−“ is the allelic effect of the negative allele (Equation (11)). “f_al+” is the frequency of the positive allele. “f_al−“ is the frequency of the negative allele.

**Table 4 ijms-24-10496-t004:** Dominance effects of the heifer conception rate (HCR) greater than the largest additive effect (δ > 1.02).

SNP	Chr	Position	Candidate Gene	δ	DR	d_DR	f_DR	DD	d_DD	f_DD	RR	d_RR	f_RR	log_10_(1/p)
*rs111023007* ^a^	4	23542497	*AGMO* (u)	4.20	12	0.535	0.147	11	−0.046	0.848	22	−7.29	0.005	60.90
*rs29023731* ^a^	4	23863959	*AGMO*	3.70	12	0.491	0.153	22	−0.044	0.841	11	−6.37	0.006	51.25
*rs136006978*	4	28692421	*TMEM196-MACC1*	1.61	12	0.378	0.256	11	−0.066	0.724	22	−2.41	0.020	28.37
*rs42269361*	4	27587323	*HDAC9*	1.62	12	0.376	0.239	22	−0.061	0.742	11	−2.42	0.019	26.87
*rs42739334* ^a^	6	102065812	*KLHL8*	3.31	12	0.376	0.116	11	−0.025	0.88	22	−5.84	0.004	26.75
*rs43480825* ^a^	6	101994654	*AFF1*	3.34	12	0.37	0.116	11	−0.024	0.88	22	−5.91	0.004	26.71
*rs43768813* ^a^	6	101887271	*AFF1*	3.35	12	0.373	0.114	22	−0.024	0.882	11	−5.93	0.004	26.46
*rs109933750* ^a^	6	102164971	*KLHL8-HSD17B13*	3.33	12	0.368	0.115	22	−0.024	0.881	11	−5.90	0.004	26.23
*rs42910197*	4	34161540	*LOC101904927*	1.91	12	0.344	0.201	22	−0.043	0.788	11	−3.09	0.011	24.65
*rs42598643*	4	24012373	*MEOX2*	1.36	12	0.311	0.244	11	−0.051	0.738	22	−2.04	0.019	19.11
*rs43377794*	4	28307693	*FERD3L-POLR1F*	1.04	12	0.288	0.294	22	−0.062	0.677	11	−1.44	0.030	16.51
*rs110248417*	8	77960968	*NTRK2*	1.62	12	0.246	0.147	11	−0.021	0.847	22	−2.73	0.007	10.88
*rs43595191*	9	49192024	*ASCC3*	1.09	12	0.221	0.217	22	−0.031	0.769	11	−1.72	0.014	9.88
*rs137681062*	8	76684305	*FRMD3*	1.41	12	0.233	0.16	22	−0.023	0.832	11	−2.32	0.008	9.78
*rs109785425*	9	48907179	*GRIK2-ASCC3*	1.06	12	0.21	0.215	11	−0.029	0.771	22	−1.66	0.014	8.82

^a^ This SNP is recommended for culling heifers with the homozygous recessive genotype based on its effect on HCR (δ > 2.0). “u” indicates the SNP is upstream of the gene. “δ” is the dominance effect of the SNP as the difference between the heterozygous dominance value and the average of the two homozygous dominance values (Equation (12)). “DR” is the heterozygous genotype with one dominant allele and one recessive allele. “d_DR” is the dominance value of the heterozygous genotype with one dominant allele (D) and one recessive allele (R) (Equation (13)). “DD” is the homozygous genotype with two dominant alleles. “d_DD” is the dominance value of the homozygous genotype with two dominant alleles (DD) (Equation (13)). “RR” is the homozygous genotype with two recessive alleles. “d_RR” is the dominance value of the homozygous genotype with two recessive alleles (RR) (Equation (13)). “f_DR” is the frequency of the heterozygous genotype. “f_DD” is the frequency of the homozygous genotype of the dominant allele. “f_RR” is the frequency of the homozygous genotype of the recessive allele.

**Table 5 ijms-24-10496-t005:** Negative impact of recessive genotypes of fifteen SNPs for heifer culling.

SNP	Chr	Candidate Gene	DPR	CCR	HCR	AFC (days)	Milk Yield (kg)	Fat Yield (kg)	Protein Yield (kg)
			Negative for fertility, age at first calving and yield traits						
*rs110035804*	1	*CCDC14*	−6.17	−8.46	−2.43	5.72	−207.53	−13.97	−9.32
*rs132847403*	1	*KALRN*	−6.01	−8.65	−2.22	5.95	−215.68	−14.63	−9.32
*rs109790791*	1	*KALRN*	−5.71	−7.95	−2.26	5.68	−196.69	−14.90	−8.84
*rs43768813* ^a^	6	*AFF1*	−5.07	−7.93	−9.12	13.25	−195.33	−11.57	−7.04
*rs43480825* ^a^	6	*AFF1*	−4.72	−7.52	−9.13	12.94	−233.31	−13.40	−8.20
*rs42739334* ^a^	6	*KLHL8*	−5.08	−7.76	−9.10	13.26	−273.91	−13.77	−9.00
*rs109933750* ^a^	6	*KLHL8-HSD17B13*	−5.00	−7.58	−9.04	12.66	−227.71	−13.04	−8.21
*rs41884737*	18	*CEBPG* (u)	−7.11	−11.18	−1.71	2.95	−119.18	−9.63	−6.48
*rs41885943*	18	*PEPD*	−6.77	−10.74	−1.65	2.66	−125.76	−9.45	−6.61
*rs133443778*	18	*PEPD*	−6.84	−10.72	−1.60	2.56	−130.03	−9.42	−6.68
*rs43746558*	18	*CHST8*	−7.23	−11.40	−1.70	2.74	−127.15	−9.74	−6.67
*rs110558219* ^b^	5	*AAAS*	−1.30	−2.01	−2.23	11.13	−737.48	−31.83	−22.73
*rs109438971* ^b^	5	*EIF4B* (d)	−1.39	−2.15	−2.25	11.23	−737.45	−32.09	−22.80
Number of cows affected by 13 SNPs			20,702 (1.73%)	17,649 (1.76%)	18,607 (1.61%)				
			Negative for fertility and age at first calving but positive for yield traits						
*rs111023007*	4	*AGMO* (u)	−2.76	−4.26	−9.21	9.35	71.09	5.28	4.19
*rs29023731*	4	*AGMO*	−2.32	−3.39	−8.23	8.40	52.49	3.95	3.49
Number of cows affected by 15 SNPs			26,254 (2.20%)	22,182 (2.22%)	25,093 (2.22%)				

DPR is the daughter pregnancy rate. CCR is the cow conception rate. HCR is the heifer conception rate. AFC is the age at first calving, where positive AFC values represent older first calving ages and are considered negative, whereas negative AFC values represent younger first calving ages and are considered positive. “u” indicates the SNP is upstream of the gene, and “d” indicates the SNP is downstream of the gene. ^a^ This Chr06 SNP was also recommended for heifer culling in the GWAS for AFC [9]. ^b^ This Chr05 SNP was recommended for heifer culling in the GWAS for AFC [9], and this study showed that this SNP also had negative effects on the three fertility traits.

**Table 6 ijms-24-10496-t006:** Comparison of additive effects from the 2023 and 2019 studies.

Trait	2023 Study (This Article)		2019 Study [8]		2019 Effects Confirmed by 2023 Study ^a^
	Number of Cows	Number of Effects	Number of Cows	Number of Effects	
DPR	1,194,736	7576	245,214	112	47 (42%)
CCR	1,001,374	3798	186,188	360	64 (18%)
HCR	1,152,219	726	269,158	15	2 (13%)

^a^ Effects from the 2019 study were compared to the top-1000 effects for DPR (Appendix A) and CCR (Appendix A) and to all 726 significant additive effects for HCR (Appendix A) from the 2023 study. DPR is the daughter pregnancy rate. CCR is the cow conception rate. HCR is the heifer conception rate.

## Data Availability

The original genotype data are owned by third parties and maintained by the Council on Dairy Cattle Breeding (CDCB). A request to the CDCB is necessary to obtain data access for this research, which may be sent to: João Dürr, CDCB Chief Executive Officer (joao.durr@cdcb.us). All other relevant data are available in the manuscript and Appendix A.

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
