# Peer review of "A Million-Cow Genome-Wide Association Study of Three Fertility Traits in U.S. Holstein Cows"

_ijms, 2023, doi:10.3390/ijms241310496_

Round 1
Reviewer 1 Report
The design of the experiment is not presented. What technology/biochips were used for cow genotyping? How were SNPs imputed. Has genotyping quality control been performed and genotyping data filtered for GWAS? Since it is necessary to take into account not only minor alleles. If all these steps have not been carried out, then the results of the analyzes will not be relevant.
Author Response
A description of cow genotyping is added, lines 578-560. CDCB has a quality control process, so the quality of the genotypes is not a concern, see link below.
https://redmine.uscdcb.com/projects/cdcb-customer-service/wiki/QC_Metrics_for_Genotyping_Laboratories
With the CDCB quality control, we could consider MAF and HWD for further SNP editing, but our method allows HWD, lines 642-644, so we only considered MAF.
Reviewer 2 Report
The authors have taken considerable efforts to compile the detailed findings of this study. Below mentioned are some of my comments:
Major concerns:
1. The Introduction portion seems to short
2. Discussion: Scientifically speaking, most portions of subheading 3.1, 3.2, 3.3 and 3.4 have contents suitable under the ‘Results’ section. Please address this issue. Especially almost all the content mentioned in subheading 3.1 and 3.2 explain the results obtained with hardly any discussion
3. The discussion needs to be improvised. The authors have taken efforts to give a detailed explanation of the results however none of its contents have been discussion in the ‘discussion’
a. One probable suggestion is explaining the functions of the most mentioned candidate gene for each trait
4. Supplementary table S1: it would be better to represent the x-axis as gene count. This is would give a better and quicker visual understanding on which pathways were most enriched
5. Gene ontology results: the authors have presented the results for biological process in the supplementary files. How about for molecular functions? Did they check this GO process? Also did they find any relevant pathway with respect to reproduction in this? If yes then kindly mention. If no then I suggest they analyze the results for molecular functions too
6. In the discussion, the authors mentioned that 15,953 SNPs in the 60K (2019) study were not found in the 80K. Did they try running the analysis upon imputing these missing 60K SNPs in 80K study?
7. Materials and methods: kindly mention the methodology adopted for gene ontology and KEGG analysis. Did the authors annotate them manually?
8. The authors seemed to have abruptly ended the discussion without highlighting the conclusion and future perspective of this study
Minor suggestions:
1. Line 251: there seems to be a typographic error. CCR was indicated instead of HCR
2. Line 368: repetition of words ‘The Chr04 example for HCR’
Author Response
Response: We thank the reviewer for a very helpful review.
The authors have taken considerable efforts to compile the detailed findings of this study. Below mentioned are some of my comments:
Major concerns:
- The Introduction portion seems to short
Answer: The intention was to focus on relevant info and background information and the writing tried to avoid distractions. This revision added a few lines, lines 59-61.
- Discussion: Scientifically speaking, most portions of subheading 3.1, 3.2, 3.3 and 3.4 have contents suitable under the ‘Results’ section. Please address this issue. Especially almost all the content mentioned in subheading 3.1 and 3.2 explain the results obtained with hardly any discussion
Answer: Agreed. The two sections of Results and Discussions are merged, and more discussions are added, e.g., section 2.6.
- The discussion needs to be improvised. The authors have taken efforts to give a detailed explanation of the results however none of its contents have been discussion in the ‘discussion’. One probable suggestion is explaining the functions of the most mentioned candidate gene for each trait
Answer: The new 2.6 was to address this comment, in addition to other places discussing the gene functions and merging Results and Discussions.
- Supplementary table S1: it would be better to represent the x-axis as gene count. This is would give a better and quicker visual understanding on which pathways were most enriched
Answer: The reviewer probably meant to say Figure S1. To address this and the next comment, we replaced the previous Figure S1 with the Current Figures S1 and S2, see response to the next comment.
- Gene ontology results: the authors have presented the results for biological process in the supplementary files. How about for molecular functions? Did they check this GO process? Also did they find any relevant pathway with respect to reproduction in this? If yes then kindly mention. If no then I suggest they analyze the results for molecular functions too
Answer: The new Figures S1 and S2 summarized the biological process, cellular component, and molecular function. For DPR, none of these directly involved reproduction. This revision added HCR, and one gene was involved in reproduction, see biological process of Figure S2.
- In the discussion, the authors mentioned that 15,953 SNPs in the 60K (2019) study were not found in the 80K. Did they try running the analysis upon imputing these missing 60K SNPs in 80K study?
Answer: The 60K is no longer used in CDCB’s genomic evaluation and would have no value to impute the removed SNPs (some may have problems), although imputing to another density should be useful for fine mapping, which is beyond the scope of this study.
- Materials and methods: kindly mention the methodology adopted for gene ontology and KEGG analysis. Did the authors annotate them manually?
Answer: A short description is added, lines 698-699, similar to the description in a Nature Genetics article, Nature Genetics 2021, 53, (8), 1250-1259
- The authors seemed to have abruptly ended the discussion without highlighting the conclusion and future perspective of this study
Answer: We share this concern. Conclusions were not included because this section was optional, and the article was already long. This revision adds this section, which should have improved the manuscript considerably.
Minor suggestions:
- Line 251: there seems to be a typographic error. CCR was indicated instead of HCR
Corrected. Thanks.
- Line 368: repetition of words ‘The Chr04 example for HCR’
Corrected. Thanks.
Round 2
Reviewer 1 Report
Missing name of chips, description of genotyping quality control, necessary filtering of genotyping dat
Author Response
Thank you.
Reviewer 2 Report
The authors have addressed all the comments placed during the 1st review report and have done a good job on revising the manuscript accordingly. I have no further suggestions.
Author Response
Chip info is provided as Table S11.
Round 3
Reviewer 1 Report
A vital step that should be part of any GWAS is the use of appropriate QC (A tutorial on conducting genome-wide association studies: Quality control and statistical analysis https://onlinelibrary.wiley.com/doi/10.1002/mpr.1608)
Without extensive QC, GWAS will not generate reliable results because raw genotype data are inherently imperfect. Errors in the data can arise for numerous reasons, for example, due to poor quality of DNA samples, poor DNA hybridization to the array, poorly performing genotype probes, and sample mix-ups or contamination. For instance, failing to thoroughly control for these data issues has led to the retraction of an article published by Sebastiani et al. (2010) in Science (Sebastiani et al., 2010, 2011; Sebastiani et al., 2012; Sebastiani et al., 2013). The results of the retracted article were affected by technical errors in the Illumina 610 array and an inadequate QC to account for those. Even though the main scientific findings remained supported after appropriate QC, the results of the new analysis deviated strongly enough for the authors to decide to retract the article.
The seven QC steps consist of filtering out of SNPs and individuals based on the following: (1) individual and SNP missingness, (2) inconsistencies in assigned and genetic sex of subjects (see sex discrepancy), (3) minor allele frequency (MAF), (4) deviations from Hardy–Weinberg equilibrium (HWE), (5) heterozygosity rate, (6) relatedness, and (7) ethnic outliers (see population stratification).
In addition, it is necessary to prune LD.
Linkage inequality (LD) is a measure of genotype correlation between a pair of variants. LD reduction are options for filtering a process so that the remaining required LD values are below a given threshold. This procedure is typically used to identify (post) an independent subset of variants used in GWAS.
Author Response
General response.
CDCB QC steps include checks at the individual and SNP levels. Individuals with call rate <90% or parent-progeny conflicts >2% were removed. X-specific SNP were also checked to validate sex information. SNPs with a departure from Hardy-Weinberg equilibrium were also removed (ref: https://www.sciencedirect.com/science/article/pii/S0022030211003079#sec0015). In addition, we applied MAF>0.05 filtering at the SNP level. Due to the MAF and HWE requirements, the minimal heterozygosity is 0.095.
Relatedness has been adjusted with our AGLS method that is equivalent to using pedigree relationships and was shown to be 40% more conservative than BOLT-LMM that uses genomic relationships based on SNPs while leaving one chromosome out. LD pruning is irrelevant here as our AGLS approach doesn't use genomic relationship matrix.
Our response to the 1st round of review provided a website that has some details of CDCB’s QC, including call rate, parent-progeny conflicts, failing of HWE.
Here are the specific answers for the 7 items.
(1) individual and SNP missingness. No individual had missing genotype because every cow was genotyped by one chip and missing genotype for any SNP was filled by imputing.
(2) inconsistencies in assigned and genetic sex of subjects (see sex discrepancy). CDCB has a control by X-specific SNPs.
(3) minor allele frequency (MAF). We required 5%.
(4) deviations from Hardy–Weinberg equilibrium (HWE). Our method allows HWD as mentioned in the 1st round of the review, but CDCB had a HWE requirement anyway, see our general response.
(5) heterozygosity rate. The MAF and HWE requirements lead to a minimal 0.095 heterozygosity.
(6) relatedness. The AGLS method is equivalent to using pedigree relationships.
(7) ethnic outliers. This is not applicable to our study.
Round 4
Reviewer 1 Report
No Comments